# The Christology of the Church of the East

**Sebastian P. Brock**

Faculty of Asian and Middle Eastern Studies, University of Oxford, Oxford OX1 2LE, UK;
sebastian.brock@ames.ox.ac.uk

**Abstract:** After setting out the background of the early history of the Church of the East, this contribution focuses on the Syriac sources of the fifth to seventh centuries which are witnesses to the development of the 'two-nature' Christology of the Church of the East, situated outside the Roman Empire during this formative period. Special attention is paid to the ambiguous term *qnoma*, which is used to render *hypostasis* in the Chalcedonian Definition, but which, for native Syriac authors, has the different sense of 'defining characteristic'. The problematic designation 'Nestorian' should be avoided since it implies completely different things to different parties. Some final thoughts are given to the ongoing significance of the tradition of the Church of the East in its various present-day manifestations.

**Keywords:** Babai; Chalcedon; Christology; Church of the East; *qnoma*; synods; Syriac

## 1. The Origins of the Church of the East

The modern representatives of the tradition of the Church of the East, namely the Assyrian Church of the East, the Ancient Church of the East, the Chaldean Catholic Church, and the Syro-Malabar Church, have adherents all over the world. In the first millennium AD, however, the Church of the East was an Asian Church par excellence: its centre of gravity was in Mesopotamia, but by the end of the first millennium and early centuries of the second millennium, its reach had extended to Central and East Asia, as well as to India. In the latter part of the fourteenth century, however, two disastrous events brought much of this to an end: the Black Death and the ravages of Timur Leng and others (Wilmshurst 2011).

The earliest history of Christianity outside and to the east of the Roman Empire is clouded in obscurity owing to a lack of reliable sources, and it is only in the first half of the fourth century that some clarity emerges with the 23 'Demonstrations' by Aphrahat and the earliest Acts of the Persian martyrs. Nevertheless, it is likely that Christianity spread beyond the eastern borders of the Roman Empire into the Parthian Empire, in what is now Iraq and Iran, the homelands of the Church of the East, at an early date. According to Acts 2:9, Jews from that region were present in Jerusalem when Peter spoke to the crowds at Pentecost. It is not unreasonable to suppose that a few of them may have carried Peter's message back with them. In any case, by c. AD 200, not only did Abercius find a Christian community in the border town of Nisibis on his travels, but the Book of the Laws of the Countries (section 46), associated with Bardaisan (d. 222), implies (section 46) that Christianity was already present to the east of the Roman Empire. By the first half of the fourth century, it is clear from the writings of Aphrahat 'the Persian Sage' that Christianity was by then well established and with an organised hierarchy. It was about this time that churches in different parts of the Roman Empire began to take an active interest in tracing their apostolic roots. One such church was that of Edessa, whose claims to apostolic origin were recorded by Eusebius in his *Ecclesiastical History* (I.13) and are found in a more developed form in the Syriac *Doctrine of Addai*, dating from the early fifth century. According to this tradition, after the Resurrection, the Apostle Thomas sent Thaddaeus (thus in Eusebius), but in Syriac Addai, to preach the Gospel. It was to

the Edessene tradition of the conversion of Edessa and its king Abgar by Addai that, at some subsequent stage, the Church in the Sasanian Empire attached their own tradition of apostolic origins through the preaching of Mari, who is described in the Life of Mari (dating from the seventh century or later) as a disciple of Addai who was sent to preach the Gospel in Mesopotamia. The link between Addai and Mari became familiarized through the title of the ancient Anaphora of Addai and Mari, whose validity as a Eucharistic prayer, despite the absence of a formal Institution Narrative, was officially recognized by the Congregation for the Doctrine of the Faith in January 2001 (Giraudo 2013).

## 2. The Development of the Christology of the Church of the East (Hainthaler et al. 2019)

It is important to recall that the Church of the East grew up outside the Roman Empire. One important consequence for the East of this is that the Church never had any part in any of the Ecumenical Councils, seeing that these were convened by the Roman Emperor and had nothing to do with bishops outside the Roman Empire. The independent development of the Church within the Sasanian Empire is clearly witnessed by the fact that it was only in 410, at a Council held in Seleucia-Ctesiphon under the auspices of a western bishop, Marutha, who was serving as an official legate of the Roman Emperor, that the Council of Nicaea (325) was 'received' by the Church in Persia. It is through the doctrinal statements of this synod and of the various subsequent synods held over the course of the fifth to the seventh century, preserved in a collection known as the *Synodicon Orientale* (Chabot 1902; English tr. in Melloni and Ishac 2023), that the development of the Christology of the Church of the East in these formative centuries can best be traced. The text of the creed adopted by the Synod of 410 was basically the Nicene-Constantinopolitan, although its actual wording in the *Synodicon Orientale* has at some point been updated, with the result that, to reach the original wording, one needs to resort to the Syrian Orthodox Acts of that synod. From their perspective, the Church of the East was 'orthodox' until the late fifth century, when they adopted the strictly dyophysite Christology of Theodore of Mopsuestia (d. 428), many of whose works had been translated into Syriac in the 430s at the famous 'Persian School' in Edessa. With the closure of the school in 489, many of its teachers, including the poet-theologian Narsai, moved across the frontier to Nisibis where, during the course of the sixth century, the School of Nisibis acted as an influential intellectual force within the Church of the East, promoting both the exegetical and Christological teaching of Theodore (Becker 2006).

Even before the closure of the Persian School of Edessa in 489, the influence of Theodore would have reached the Church in the Sasanian Empire by way of students returning from Edessa. Thus, it is no surprise that the Christological statement of the Synod of Seleucia-Ctesiphon of 486 employs language that clearly belongs to the Antiochene Christological position (Brock 1992b):

> Let our faith in the dispensation of Christ be in the confession of the two natures, of the divinity and of the humanity, while none of us shall dare to introduce mixture, mingling or confusion into the differences of these two natures; rather, while the divinity remains preserved in what belongs to it, and the humanity in what belongs to it, it is to a single Lordship and to a single (object of) worship that we gather together the exemplars of these two natures because of the perfect and inseparable conjunction that has occurred for the divinity with respect to the humanity. And if someone considers, or teaches others, that suffering and change have attached to the divinity of our Lord, and if he does not preserve, with respect to the union of the prosopon of our Saviour, a confession of perfect God and perfect Man, let such a person be anathema.

While the formulation makes use of the terms 'nature' and 'person' (using the Greek loanword *prosopon*), as found in the Chalcedonian Definition of Faith, issued thirty-five years earlier, there is no mention of the term *qnoma*, which is the standard rendering of Greek *hypostasis* and whose ambiguity in Syriac later caused misunderstanding and trouble. The hostility towards mixing language is shared with Chalcedon, and the terms

'union' and 'Perfect God and Perfect Man' would have been widely in use. 'Conjunction' (corresponding to Greek *sunapheia*), however, and the opposition to Theopaschite language, are distinctive of the wider Antiochene tradition in general.

From synods of the Church of the East in the sixth century, there survive Christological statements from the years 544, 554, 576, 585, and 598 (Brock 1992b; Ebeid 2016; Metselaar 2019). It is striking to note that, while the Chalcedonian terms 'nature' and 'property' sometimes feature, *qnoma* is only used in connection with the Trinity (of three *qnome*) and never with reference to Christ. The same applies for the Synod of 605, which specified 'each of us should receive and accept all the commentaries and writings of the blessed Theodore the Interpreter'.

The first time that reference is made to two *qnome* in Christ is in the report (in the *Synodicon Orientale*) of a public dispute, held in 612, between representatives of the Church of the East and the 'Severan Theopaschites', that is, Syrian Orthodox followers of Severus of Antioch, who at one point ask 'Is it the Nestorians or the [Syrian Orthodox] monks who have turned aside from the foundations of the faith transmitted by the teachers of old?' They continue with their attack by asking 'Previous to Nestorius, is there anyone who says that Christ is two natures and two *qnome*?' From the reply of the representatives of the Church of the East, it is clear that their understanding of *qnoma* is different from the Chalcedonian use of the term *hypostasis* (Chabot 1902, p. 575):

> It is clearly apparent that Christ is perfect God and perfect Man. Now, he is said to be God, being perfect in the nature and *qnoma* of divinity, and he is then said to be perfect Man, being perfect in the nature and *qnoma* of humanity. And just as it is made known from the opposition (expressed in) the words just used, that Christ is two natures and two *qnome*, so too, from the fact that they refer to the one Christ, Son of God, it is made known that Christ is one—not in the oneness of nature and of *qnoma*, but in a single *prosopon* of Sonship and a single authority, a single governance, and a single Lordship.

In this passage, *qnoma* has the sense of 'defining characteristic', and definitely does not have the same sense as *hypostasis* in the Chalcedonian Definition. To obtain a much more detailed understanding of the Church of the East's usage, one needs to consult the 'Book of the Union [*sc.* of the two Natures]' by Babai the Great (d. 628; Chediath 1982), a work which has remained authoritative until the present day, along with the much later work entitled 'the Pearl (*marganita*)' by 'Abdisho' of Nisibis (d.1318). It is indeed possible that Babai himself was present at the Disputation of 612, being a prominent figure in the Church at the time. It is significant that in his treatise, when referring to the *qnome* in a Christological context, Babai usually speaks of 'the two natures and *their qnome*', which brings out this sense of the term as 'distinctive characteristic'; moreover, on a number of occasions, Babai emphasizes that, in Christ, these two *qnome* have been united ever since the moment of conception.

Given that *qnoma* was also the standard Syriac rendering of Greek *hypostasis*, it is easy to see how confusion and misunderstanding could arise. Although 'two *qnome*' are not attested in East Syriac sources until the early seventh century, it is likely that Babai's understanding of the terms goes back to the early sixth century when writers of the opposite Christological tradition, such as Jacob of Serugh (d. 521) in his Letter 16, express alarm at hearing talk of 'two *qnome*' which they would have understood as 'two *hypostaseis*'. Conversely, East Syriac theologians, on hearing of Chalcedon's 'one *hypostasis*', rendered into Syriac as *qnoma*, would have been equally horrified, given their own different understanding of *qnoma*, a reaction which can be found reflected in a letter on Christology by Patriarch Isho'yahb II (628–646) from a little over a century later.

Not unconnected with the two different understandings of *qnoma* is the difference in understanding of the term 'nature' in the Chalcedonian Definition by the opposing sides: for the East Syriac tradition, 'nature' (*kyana*) was close in sense to *ousia*, 'essence', whereas in the West Syriac tradition, it was almost synonymous with *hypostasis*. Given these different understandings, it is not surprising that both the Church of the East and the

Syrian Orthodox found the Chalcedonian Definition illogical, though for different reasons. As Isho'yahb II expressed it (Sako 1983, pp. 146–7 [tr.], 170 [text]):

> Although those who gathered at the Synod of Chalcedon were clothed in the intention to restore the faith, yet they too fell away from the true faith: owing to their feeble phraseology they provided a stumbling-block for many. Although in accordance with the opinion of their own kinds, they preserved the true faith with the confession of the two natures, yet by their formula of the one *qnoma* (*hypostasis*), it seems they tempted weak minds. As a result, a contradiction occurred, for with the formula 'one *qnoma*' they corrupted the confession of the two natures, and with the 'two natures' they rebuked and refuted the 'one *qnoma*'.

It was above all these two terms, *kyana* and *qnoma*, with their ambiguity, which led to confusion and bitter controversy in Late Antiquity, with no side making the intellectual effort to understand the opposing sides' real position and their different understanding of the technical terms involved. In this connection, it is instructive to observe what happened when a liturgical verse text, 'Blessed is the Compassionate One', attributed to Babai the Great, was taken over into Maronite tradition. Towards the end, the East Syriac version has:

> The natures {sc. divinity and humanity) are preserved with their *qnome* in a single *prosopon* of a single Sonship.

The Maronite version alters the phrase 'with their *qnome*', to them unfamiliar, by substituting 'without confusion'. The Chaldean Catholic editor of the East Syriac hymn, Paul Bedjan, correctly understood the meaning of 'with their *qnome*', but realizing that *qnome* would be misunderstood, substituted 'with their properties', capturing the right sense. Furthermore, to adapt the phraseology more clearly to Chalcedonian terminology, he also altered 'in a single prosopon' to 'in a single *qnoma*' (in the sense of *hypostasis*). In modern times, yet further confusion has been added by the regrettable, but quite frequent, translation of *qnoma* in a Christological context as 'person', giving the totally wrong impression that the Church of the East professed 'two persons' in the incarnate Christ, whereas from the first, the East Syriac tradition has emphatically refuted the charge that they believed in two Sons, the Son of God and the son of Mary.

With the advent of Islam and, in due course, the adoption by all the Christian communities of Arabic for most theological writing, suitable equivalents in Arabic had to be found which would serve not only the requirements of intra-Christian debate and controversy but also meet the needs of defending Christian doctrines in the face of Muslim critiques (Ebeid 2018).

In much of Western scholarship, the Chalcedonian Definition has been treated as the yardstick against which other positions were judged. In its most simplistic (but widespread) form, Chalcedon provided the via media between two 'heretical' positions, one of which was that of the Church of the East. A closer look at the evidence, however, indicates that the situation was far more complex, and it would be considerably more helpful (and especially so in an ecumenical context) to work with a sevenfold model that indicates much more satisfactorily the variety of positions within the Christological spectrum and the close relationship in many respects of the Church of the East with part of the Chalcedonian tradition. At either end of the spectrum are two symbolic names of persons whose real views remain far from clear: at the Alexandrine end, there is 'Eutyches', whose alleged position should be rejected by everyone, while at the other end is 'Nestorius', as seen by his enemies. In between, there are five different positions, all of which hold that the incarnate Christ is consubstantial both with his Father and with us, and all of which can be seen as 'orthodox' from an ecumenical standpoint. Reading, as it were from the Alexandrine end to the Antiochene, these five positions are represented by: 1, the Syrian Orthodox (and other Oriental Orthodox); 2, the way of silence (promoted by the Emperor Zeno's *Henoticon* of 482 and represented especially by Jacob of Serugh and the author of the Dionysian corpus); 3, the Neo-Chalcedonians of the early sixth century; 4, other Chalcedonians (including Rome); and 5, the Church of the East. Each of these five positions has its own particular concerns

and emphases, several of which are shared by two positions. Thus, many concerns are shared between the Syrian Orthodox and the Neo-Chalcedonians on the one hand, and on the other, between Chalcedonians (such as Theodoret and Pope Leo) and the Church of the East. For the latter pairing, one thinks especially of their common dislike of Theopaschite language and of the *communicatio idiomatum.*

The analytical Christology of the theological élites which resulted in the three-way split in Eastern Christianity had an important counterpart on a different level: the Christological language of the liturgy, where the adage *lex orandi lex credendi* applies. The Chalcedonian technical terms only rarely turn up in the extensive liturgical texts of the Church of the East. What marks these texts as distinctive is the preference for certain specific imagery in the context of Christology. In some instances, this was once imagery widely shared with other traditions, going back to the New Testament, but which, in the course of the Christological controversies, was dropped by other traditions and only preserved in the Church of the East. A notable example concerns the imagery of Christ's body as a temple: this has an excellent basis in the Gospel of John (2:9) where Jesus challenges the Jewish authorities with the word 'Destroy this temple, and in three days I will rebuild it'. In East Syriac liturgical texts, the image of Christ's body as a temple is often combined with the idea of 'indwelling'; thus, in the Hudra at the season of the Nativity, the two images are juxtaposed:

> The daughter of David has given birth to the Child of Wonder (cf. Isaiah 9:6). Christ: (at once) the Holy of Holies and the Power of the Most High; the Temple and the One who took it; the dwelling place and the One who dwells in it—one *prosopon* (with) two likenesses. (*Hudra*, ed. T. Darmo, I, pp. 171–2)

The situation with one of the earliest phrases by which the Syriac-speaking Church referred to the incarnation, namely 'he put on the body' (*lbesh pagra*), is instructive. Early Syriac writers such as Ephrem employed an abundance of phrases involving *lbesh*; thus, one finds 'he put on Adam, humanity, our humanity', 'our body', 'our likeness', etc. (Brock 1992a). Sometimes, too, Christ's body would be described as his 'garment'; this occurs occasionally in Ephrem (e.g., *Hymns on the Nativity* III.20), but in the Christological controversies of the fifth century, 'clothing' imagery fell out of favour with Syriac writers in the Alexandrine tradition of Christology such as (and in particular) Philoxenos of Mabbug (d. 523), who regarded it as promoting a diphysite position. In the East Syriac tradition, however, it was actually extended, in view of its implicit support for a two-nature Christology. It is noticeable that references to Christ's body as a robe (*estla*, from Greek *stolē*) and 'purple' are only to be found in East Syriac authors (Brock 2022); thus, for example, the poet Narsai (d. c.500) speaks of 'the robe of (Christ's) bodily state' (ed. Mingana, II, p. 161). The imagery of Christ's body as his 'purple (robe)' is first attested in the *Doctrina Addai*, dating probably from the 420s; in the course of a sermon, Addai is represented as saying '(Christ's) body is the pure purple (robe) of His glorious divinity by which we are able to see His hidden Lordship'. At the Second Council of Ephesus (449), the bishop of Edessa, Ibas, was denounced by some of his clergy for his alleged use of this image. It reappears in Narsai who, in Homily 56, wrote 'I confess the King who put on the purple of the body of Adam', using wording that goes back to Diodore of Tarsus.

Despite the fact that the imagery of Christ's body as a temple, or as a garment, were so popular in East Syriac theological writing, it was not entirely dropped in the West Syriac tradition where it can occasionally be found in liturgical poetry, a notable example being in the Mosul edition of the *Fenqitho*, or Festal Hymnary (III, p. 231):

> 'Blessed is He whose garment was our body, and our frame (*gushman*) a temple for His Being'.

Towards the end of his *Book of the Union*, Babai points out that each of the various metaphors and analogies used in connection with the incarnation points to a particular aspect of how to understand the 'union' of the two natures; thus, for example, clothing imagery points to the voluntary nature of the union. Accordingly, it is important to employ

a variety of images in order to provide a balanced picture. Interestingly, exactly the same point was made a century earlier by Philoxenos.

## 3. Some Key Christological Insights

Underlying the East Syriac position on Christology is a concern to bring out the transcendence of the Godhead by emphasizing the 'chasm' (as Ephrem expressed it) between God and his creation, between divinity and humanity. This concern is reflected clearly in the distinction which is strictly maintained between the two natures in Christ and the dislike of Theopaschite language and of the transfer of the properties between the divinity and humanity in Christ. Linked with this concern is the difficulty felt with the wording of John 1:14 ('The Word became flesh') and the suspicion that 'becoming' involved 'change'; this hesitation gave rise to a different understanding of the syntax found in Babai (Chediath 1982, pp. 96–7) but going back earlier: 'As for the Word, flesh came into being, and He dwelt in us', where 'us' is understood as 'one of us' or 'our human nature'. 'Abdisho', in his *Pearl* (III.1), accords to this interpretation the authority of 'a voice from heaven'.

The distinction between the two natures is in fact also essential for the underlying East Syriac concept of how Christ, the incarnate manifestation of God the Word, effects salvation for humanity. This is understood as having been brought about through Christ's own humanity which is then raised up and 'deified' at the Ascension together, potentially, with that of all human beings (cf. Eph. 2:6). This 'soteriology from below' is in contrast to the conceptual model implicit in the miaphysite tradition, according to whose 'soteriology from above' it is at the conception of Christ that salvation is brought about, the moment when God the Word, by becoming a human being, underpins all humanity. With these two contrasting viewpoints in mind, of how salvation comes about, it is easy to see why each side thought that the other side's approach completely undermined the process by which salvation came about. Needless to say, it is not a question of either/or, but of both/and, as indeed the liturgical traditions of each side were in fact well aware and on occasion brought out.

The same concern for the transcendence of the Godhead and the need to maintain the distinction between the two natures of the incarnate Christ also explains why the Church of the East has preferred the title 'bearer of Christ', rather than 'bearer of God (*theotokos*)' for Mary. While Babai the Great, for example, conceded that in certain contexts, it is permissible to use 'bearer of God', he stresses that it is more important to keep in mind the co-existence of the two natures in Christ, the incarnate Word, in view of the final raising up of his humanity—seen as essentially *our* humanity, at the Ascension. It is important to note that the Church of the East's dislike of the title 'bearer of God' did not prevent it from developing a rich collection of types and symbols of Mary.

A further aspect of the 'wonder' (Is. 9:6) of Christ, the Word made flesh, is brought out in the frequently found statement that Christ is 'a member of our (human) race (*bar gensan*)', thus lending emphasis to his solidarity with human beings and so a source for legitimate human pride.

## 4. The Term 'Nestorian'

The term 'Nestorian', alluding to Nestorius, bishop of Constantinople, deposed at the Council of Ephesus, was first used by those hostile to the Christology of the Church of the East during the bitter controversies of the fifth and sixth century, but by the time of the Middle Ages and under Islamic rule, it had also come to be used as a term of self-definition by writers of the Church of the East, and this usage continued into the twentieth century. Although the name 'Nestorian' is still upheld by some today, in the context of ecumenical relations, it is highly problematic, since it can be (and is) the source of serious misunderstandings; accordingly, the term is much better avoided (Brock 1996). The basic problem lies in the fact that the name 'Nestorius' conjures up at least three completely different connotations for different people: from the viewpoint of the Church of the East, Nestorius was a Greek theologian who upheld the 'two-nature' Christology in the face of



Cyril of Alexandria. In contrast to the case of Theodore of Mopsuestia, Nestorius in fact had very little influence on the Christology of the Church of the East; his writings were very little known, apart from the *Liber Heracleidis*, a late work which was only translated into Syriac in the sixth century, and he was chiefly revered as the putative author of one of the three eucharistic anaphoras in use. For the Chalcedonian Church, however, Nestorius was a heretic who had been deposed at the Council of Ephesus, largely in connection with his disapproval of Mary's description as *Theotokos*. For the Oriental Orthodox Churches, Nestorius was the arch-heretic whose strict dyophysite Christology was abhorrent to their miaphysite, or one-nature, standpoint. Finally, for modern scholarship, what Nestorius really thought and taught remains a matter of uncertainty and dispute. Given such a variety of contradictory understandings of the term 'Nestorian', in almost all contexts, it is best not to employ it.

### 5. The Ongoing Significance of the Tradition of the Church of the East

As was pointed out at the outset, the tradition of the Church of the East is represented by four different ecclesial bodies, two non-Catholic and two Catholic. In the case of the two non-Catholic Churches, the Assyrian Church of the East and the Ancient Church of the East, the split only goes back to 1964, whereas the two Catholic branches, the Chaldean Catholic Church based in the Middle East and the Syro-Malabar Church in India, trace their origins back to the sixteenth century. The history of the Syriac Churches in India is highly complex and the terminology confusing. This applies in particular to the term 'Chaldean' which, in India, has no connection with the Chaldean Catholic Church of the Middle East but instead designates a part of the Assyrian Church of the East.

Ecumenical dialogue between the Chalcedonian and non-Chalcedonian Churches only commenced in the latter half of the twentieth century, and at first, in the 1960s and 1970s, it was only the Oriental Orthodox Churches which were involved, with separate Eastern Orthodox/Oriental Orthodox and Catholic/Oriental Orthodox dialogue on both non-Official and Official levels. The year 1984 witnessed a significant new development with an official meeting between the Catholicos Patriarch Mar Dinkha IV and Pope John Paul II. The following year, Mar Dinkha applied to the Middle East Council of Churches for the membership of the Assyrian Church of the East, but this was turned down, as were several successive applications, largely due to a misunderstanding of the Assyrian Church of the East's Christology and the opposition of the Coptic Orthodox Church. Sadly, this situation continues to the present day, a glaring ecumenical wound which cries out for healing.

An important initiative was taken in 1994 by the non-Official Foundation PRO ORIENTE, based in Vienna; this was the inception of a series of meetings entitled 'Syriac Dialogue' and involving for the first time all the different Syriac Churches, with their verbally clashing Christological formulas (PRO ORIENTE 1994–1998). These meetings provided an ideal forum for dispassionate discussion and for the realisation that, underlying the conflicting wording of the different formulas, there lay a shared understanding of the mystery of the incarnation: provided each side made the effort to understand what the other side was really saying, as opposed to what they were imagined to be saying, real progress could be made, and traditional misconceptions removed.

It was thanks to these and similar developments that it was possible for Mar Dinkha IV and Pope John Paul II to make a Common Declaration of Faith in November 1994. This includes the following (Brock 2004, pp. 54–55):

> ... our Lord Jesus Christ is true God and true man, perfect in His divinity and perfect in His humanity, consubstantial with the Father and consubstantial with us in all things but sin. His divinity and His humanity are united in one person, without confusion or change, without division or separation. In Him has been preserved the difference of the natures of divinity and humanity, with all their properties, faculties and operations. But far from constituting 'one and another',

the divinity and the humanity are united in the person of the same and unique
Son of God and Lord Jesus Christ, who is the object of a single adoration.

The twentieth century has been a disastrous one for all the Middle Eastern Churches. Although the genocide during the First World War is primarily associated with the Armenians, the Syriac Churches were equally affected, and subsequent political events over the course of the twentieth century and into the twenty-first have resulted in completely new challenges. Perhaps chief among these has been the progressive emptying of the Middle East of its Christian population and the creation of diasporas scattered over all five continents. This means that the tradition of the Church of the East has become visibly a part of world Christianity.

While emigration on a large scale has left the homelands vastly depleted, it has, nevertheless, brought about new possibilities, not least in the field of higher education, facilitating (among other things) the transition into the digital world. The use of modern technology, especially as developed by younger lay members of the Syriac Churches, has made available in digital form liturgical texts and services that can be accessed almost anywhere with great ease. This has a double benefit: on the one hand, it can open up new possibilities for catechetical purposes and for teaching the theology and history of the individual Churches; at the same time, in a wider ecumenical context, it can make available to people of other Churches a path of access to the rich Syriac liturgical and musical traditions.

Finally, mention should be made of a particular context in which the tradition of the Church of the East can play an important role. It is a remarkable fact that one of the main growth areas of Christianity today is in China. It would seem of great importance that this development should make a connection with the past history of Syriac Christianity in China in the Tang and Yuan periods. An important initiative in helping to make these connections at an academic level has been taken in Salzburg with a series of conferences, commencing in 2003, and their associated publications. In the modern context, the fact that the tradition of the Church of the East can claim to represent a truly indigenous Asian form of Christianity is of no small significance.

**Funding:** This research received no external funding.

**Institutional Review Board Statement:** Not applicable.

**Informed Consent Statement:** Not applicable.

**Data Availability Statement:** No new data were created or analyzed in this study. Data sharing is not applicable to this article.

**Conflicts of Interest:** The author declares no conflict of interest.

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
