# Peer review of "The Christology of the Church of the East"

_religions, doi:10.3390/rel15040457_

Round 1

Reviewer 1 Report

Comments and Suggestions for Authors

Against the background of the history of the Church of the East, and particularly its development in the Sasanian Empire in the 5th to 7th centuries, the author traces the development of the dyophysite christology of the Church of the East, especially in light of the Syriac understanding of qnoma (defining or distinctive characteristics) in counter distinction to the Chalcedonian understanding of hypostasis (person), and the confusion that occurred in the East from translating one term by the other.  The authors draws out key christological insights from this history and underscores the ongoing significance of the legacy of the Church of the East.  This is a clear, coherent, instructive account of both the historical significance and contemporary relevance of the Church of the East.  There are only some minor corrections in grammar, spelling, and formatting needed. 

Comments on the Quality of English Language

The author should use a period instead of a semi-colon in lines 17, 31, and 309,

The author should use a period instead of a colon in lines 69, 153, 205, 306, 327, and 381.

In line 67, "be updated" should be changed to "been updated."

In line 123, "if" should be changed to "of."

Also, in the Abstract:  change "Easr" to "East," change "Syrisc" to "Syriac," and change "witsesses" to "witnesses."

Author Response

Thank you very much for catching these things

Reviewer 2 Report

Comments and Suggestions for Authors

This paper makes a connection between the tradition of the Church of the East, with four churches seeing themselves as a continuation of this tradition, and the development of the Christian faith in Asia, especially China. This gives the exposition an interesting relevance, not least because it is made clear that a) the Church of the East developed independently of and outside the Roman Empire, and b) that the association of the Church of the East with Patriarch Nestorius should not mean that the Christological doctrine of two natures being espoused is substantially different from that of the Council of Chalcedon (451).

The paper outlines the historical origins, the development of a distinctive Christology with its own emphases, rejecting the term "Nestorian" as too ambiguous (three different connotations), and concludes with a cogent reflection on the significance of this Eastern ecclesial tradition, both ecumenically and in terms of Christian mission.

It seems to me that this contribution does not put forward any substantially new insights, as also evidenced by its reliance on earlier work by Sebastian Brock in particular, and by the agreement expressed with the Roman Catholic Church. The contribution is well and clearly written, with a clear structure and, as mentioned, an interesting modern relevance. The remark at the end of p. 5 concerning the interpretation of clothing imagery, stressing the voluntary nature of the union, is very interesting. The second section of p. 6 could also employ the distinction between a soteriology from above and one from below. There are also other publications with the same title; perhaps a slight alteration is advisable.

Some editorial issues:

- the contribution contains a huge number of double spaces, which should be removed.

- the contribution contains a number of quotations that are not sufficiently recognizable marked; this should be changed.

- Typing errors:

                - p. 1, r. 3: Easr, should be East

                - p. 1, r. 4: Syrisc, should be Syriac

                - p. 2, r. 85: in the quotation a phrase is missing: “remains preserved in what belongs to it, and the humanity in what belongs to it, it is to a single Lordship”

                - p. 2, r. 86: to, should be two (“these two natures”)

                - p. 3, r. 101: Metselaar 2019 is nog covered in the bibliography (Defining Christ: The Church of the East and nascent Islam)

                - p. 3, r. 123: has the sense if, should be has the sense of

                - p. 3, r. 123: and definitely not have, should be and definitely not has

                - p. 3, r. 124: the same as hypostasis, should be the same sense as hypostasis

                - p. 4, r. 191: within of the Christological, should be within the Christological

                - p. 4, r. 195: rejected by everyone), should be rejected by everyone

                - p. 5, r. 252: there is a quotation mark at the end, but not at the beginning of the phrase

                - p. 6, r. 265: difficulties, should be difficulty

Author Response

Thank you very much for catching all these things which my aging eyesight missed; they will all be seen to